# Role of Bronchoscopic Techniques in the Diagnosis of Thoracic Sarcoidosis

**DOI:** 10.3390/jcm8091327

**Published:** 2019-08-28

**Authors:** Cecília Pedro, Natália Melo, Hélder Novais e Bastos, Adriana Magalhães, Gabriela Fernandes, Natália Martins, António Morais, Patrícia Caetano Mota

**Affiliations:** 1Faculty of Medicine, University of Porto, Alameda Prof Hernâni Monteiro, 4200-319 Porto, Portugal; 2Department of Pulmonology, Centro Hospitalar Universitário de São João, Alameda Prof Hernâni Monteiro, 4200-319 Porto, Portugal; 3Institute for Research and Innovation in Health (i3S), University of Porto, Rua Alfredo Allen 208, 4200-319 Porto, Portugal

**Keywords:** sarcoidosis, diagnosis, standard bronchoscopic techniques, recent bronchoscopic techniques

## Abstract

The diagnosis of sarcoidosis relies on clinical and radiological presentation, evidence of non-caseating granulomas in histopathology and exclusion of alternative causes of granulomatous inflammation. Currently, a proper diagnosis, with a high level of confidence, is considered as key to the appropriate diagnosis and management of the disease. In this sense, this review aims to provide a brief overview on the role of bronchoscopy in the diagnosis of thoracic sarcoidosis, incorporating newer techniques to establish, including endobronchial ultrasound-guided transbronchial needle aspiration (EBUS-TBNA), endoscopic ultrasound-guided fine-needle aspiration (EUS-FNA), transesophageal ultrasound-guided needle aspiration with the use of an echo bronchoscope (EUS-B-FNA) and transbronchial lung cryobiopsy (TBLC). Most of the literature reports the diagnostic superiority of endosonographic techniques, such as EBUS-TBNA alone or in combination with EUS-FNA, over conventional bronchoscopic modalities in diagnosing Scadding stages I and II of the disease. Moreover, TBLC may be considered a useful and safe diagnostic tool for thoracic sarcoidosis, overcoming some limitations of transbronchial lung biopsy (TBLB), avoiding more invasive modalities and being complementary to endosonographic procedures such as EBUS-TBNA.

## 1. Introduction

Sarcoidosis is defined as a multisystem granulomatous disorder of unknown etiology, first described in 1869 by Jonathan Hutchinson [1]. It is a worldwide disease that usually affects patients of working age, with a prevalence of about 4.7–64/100,000 inhabitants and an incidence of 1.0–35.5/100,000 per year [2]. There are no general data from Portugal, although, in 2017, Cardoso et al. [3] published the first report of cases of sarcoidosis in the north of Portugal and the epidemiological characteristics reported were similar to those of other Western European populations.

With regards to clinical features, sarcoidosis has variable clinical manifestations and prognosis. Although it can affect any organ, pulmonary and mediastinal lymph node involvement appears in 90% and 85% of cases, respectively [4]. These data are in line with the prevailing pathogenic hypothesis for sarcoidosis, suggesting that it probably requires exposure to more than one exogenous antigen in genetically susceptible hosts, contributing to distinct phenotypes in different ethnicities. On the other hand, although the immunopathogenesis of sarcoidosis is not fully understood, it seems to imply the relationship between the antigen, the human leucocyte antigen (HLA) class II molecules and T-cell receptors. To date, several environmental and microbial agents have been suspected, including possible mycobacteria and propionibacteria [4].

Sarcoidosis’ diagnosis is based on three basic criteria: (1) clinical and radiological presentation, (2) evidence of non-caseating granulomas in histopathology and (3) no alternative disease associated with granulomas, such as tuberculosis, sarcoid-like reactions in cancers and lymphomas [4,5].

Clinical manifestations depend on several factors, such as ethnicity, gender, age, location and extension of organ involvement [6], while radiological presentation is based on chest X-ray findings, namely pulmonary parenchymal and hilar lymph node abnormalities. Scadding staging classifies thoracic sarcoidosis into the following stages: stage 0 (no adenopathies, pulmonary infiltrates or fibrosis), stage I (hilar adenopathies alone), stage II (hilar adenopathies and infiltrates), stage III (infiltrates alone) and stage IV (advanced fibrosis) [7,8,9]. Although in some typical situations, such as Loëfgren syndrome, a presumptive diagnosis may be based on clinicoradiographic findings alone, in most cases, the histological finding of non-caseating granulomas remains critical in diagnosing sarcoidosis. Moreover, a diagnosis of high confidence level is the key to an appropriate clinical and therapeutic management of the disease, due to its heterogeneous evolution [10]. Thus, the diagnostic procedure of choice depends on a clinician’s pre-test probability for the diagnosis of sarcoidosis and the Scadding radiographic stage of the disease [11].

There are several procedures for obtaining tissue for histological evaluation. Bronchoscopy is frequently performed, as it allows tissue sampling from several anatomic sources potentially affected by the disease, such as airways, lung parenchyma and hilar/mediastinal lymph nodes. The most available are standard bronchoscopic methods, such as endobronchial biopsy (EBB), transbronchial lung biopsy (TBLB) and conventional transbronchial needle aspiration (cTBNA) [11,12]. Bronchoalveolar lavage fluid (BALF), even though is not a means to acquire tissue for histological evaluation, is considered a standard technique in the diagnostic work-up of diffuse parenchymal lung diseases (DPLD), including sarcoidosis [5,13]. In the last decade, new techniques of endoscopic ultrasonography, such as endobronchial ultrasound-guided transbronchial needle aspiration (EBUS-TBNA), endoscopic ultrasound-guided fine needle aspiration (EUS-FNA) or transesophageal ultrasound-guided needle aspiration with the use of an echo bronchoscope (EUS-B-FNA), were widely incorporated into clinical practice [11,12]. Also, transbronchial lung cryobiopsy (TBLC), successfully used in the diagnosis of other DPLD and a valid alternative to surgical lung biopsy in many situations, is a promising new technique that may have a role in diagnosing pulmonary sarcoidosis [14]. Table 1 briefly summarizes both standard and recent bronchoscopic modalities for diagnosing sarcoidosis with a series of diagnostic yields estimated in the literature.

## 2. Standard Bronchoscopic Techniques

Flexible bronchoscopy has revolutionized the diagnosis of sarcoidosis, changing from more invasive procedures (e.g., rigid bronchoscopy, mediastinoscopy and surgical lung biopsy) to less invasive ones (e.g., EBB, TBLB and cTBNA). In this review, BALF was also considered as a standard bronchoscopic technique.

### 2.1. Endobronchial Biopsy (EBB)

EBB is an established minimally invasive outpatient technique, enabling sampling, under direct vision of the proximal airways [35]. It is particularly useful in patients with endobronchial sarcoidosis who may experience cough, wheezing, stridor or signs of major airway obstruction [16].

The diagnostic yield of EBB for sarcoidosis ranges from 20% to 61% [15,16]. It increases in cases of macroscopically mucosal abnormalities, such as cobblestone appearance (54–91%); however, in cases of normal-appearing mucosa, EBB may also show granulomas in histology (20–40%) [35]. In addition, the diagnostic yield also appears to improve with further biopsies and it is advocated that at least three samples from different sites, such as main carina, subsegmental carina and/or endobronchial nodules should be taken, even in the absence of typical sarcoidosis characteristics [16]. Nonetheless, EBB has a lower diagnostic yield than TBLB. It is usually recommended as an additional procedure to TBLB, even when endobronchial disease is not evident [35].

Despite the combined used of TBLB and EBB, about one-third of all bronchoscopies fail in diagnosing sarcoidosis [36]. Finally, although EBB has a very low risk, it may present the following potential complications: extensive cough, intolerance and minor bleeding [35].

### 2.2. Transbronchial Lung Biopsy (TBLB)

As reported in the ATS/ERS/WASOG Statement on sarcoidosis, published in 1999, TBLB was considered the first choice for tissue sampling diagnosis [5]. The diagnostic yield of TBLB for suspected sarcoidosis ranges from 37% to 90%. As expected, it varies according to the operator’s experience and radiological Scadding stages, being higher in stages II and III (stage I: 66%; stage II: 80%; stage III: 83%) [16,17,18,19]. Benzaquen et al. [16] recommended that at least four samples should be obtained in all patients through TBLB, regardless of their radiological stage.

As limitations of this technique, we highlight those related to the sample obtained, namely (1) quality, since it cannot represent the lung architecture due to the crush effect of TBLB and (2) small size [16]. On the other hand, the possible complications of TBLB include hemoptysis (4%) and pneumothorax (2%) [37]. However, with the introduction of new endoscopic methods for diagnosing sarcoidosis, TBLB has decreased in its relevance over the last decade [16].

### 2.3. Conventional Transbronchial Needle Aspiration (cTBNA)

TBLB and EBB have long been the only endoscopic techniques available to diagnose sarcoidosis, until cTBNA enabled bronchoscopists to sample mediastinal lymph nodes with higher diagnostic yields and lower adverse events [38].

Conventional TBNA, first performed with a rigid bronchoscope and then with a flexible bronchoscope, is a well-established technique, but not an image-guided procedure, and, therefore, its diagnostic yield varies widely, ranging from 6% to 90% with better results in stage I of the disease [16,19,20,21]. This variability in diagnostic yield appears to be related to both procedural and clinical factors, such as operator’s skills, lymph node size, localization and underlying disease [23,38,39].

Combination modalities, such as cTBNA, EBB and TBLB may increase the yield to 70–90%, but it is often associated with longer procedure length and more complications [40,41,42]. Indeed, although cTBNA has been used for more than three decades and is widely described for both malignant and non-malignant diseases, because of high variations in diagnostic yield and unconfirmed safety concerns, it remains an underused technique [43]. Currently, cTBNA is being progressively replaced by the implementation of endobronchial ultrasound (EBUS) (described below) in most specialized centers.

### 2.4. Bronchoalveolar Lavage Fluid (BALF)

BALF is the simplest and least invasive bronchoscopic procedure and is routinely recommended as an additional procedure to EBB and TBLB in the diagnostic work-up of patients with sarcoidosis [16]. BALF should be obtained at the beginning of the bronchoscopic procedure to avoid blood contamination [16]. Several authors concluded that BALF lymphocytosis and a CD4/CD8 ratio ≥3.5 support the diagnosis of sarcoidosis in patients with a typical clinical picture, obviating the need for further biopsy confirmation, although 10% to 15% of cases have a normal cell count [13,44,45,46]. These studies demonstrated that a CD4/CD8 ratio above 3.5 shows a high specificity of 93–96% for sarcoidosis, although the sensitivity is low (around 52–59%) [13,44,45,47,48]. Kantrow et al. [49] doubt the clinical utility of this ratio on the basis of the observation that is highly variable [13]. These authors found that only 42% of 86 patients with biopsy-proven sarcoidosis had a BALF ratio greater than 4.0, and that 12% even had an inverted ratio below 1.0, reflecting a predominance of CD8^+^ T-lymphocytes [49].

In this context, the differential fluid cells profile may also be useful in distinguishing sarcoidosis from the most common DPLD. Considering that integrin CD103 is expressed by lung lymphocytes, Mota et al. found that patients with sarcoidosis had a significantly reduced CD103 expression in BALF T-lymphocytes, which was more pronounced in the CD4 subset. The BALF CD103^+^CD4^+^/CD4^+^ ratio below a cutoff point of 0.45 was associated with a better diagnostic performance for sarcoidosis, even in cases with a CD4^+^/CD8^+^ ratio < 3.5 [13]. Thus, the evaluation of CD103 expression in BALF CD4^+^ T lymphocytes may be a reliable tool in the sarcoidosis diagnostic work-up, regardless of the CD4^+^/CD8^+^ ratio. In addition, BALF study also has some importance in the differential diagnosis of sarcoidosis, as it can exclude granulomatous infections, such as fungal or mycobacterial infections [16].

## 3. Recent Bronchoscopic Techniques

In the past decade, the diagnostic approach of sarcoidosis has been revolutionized by the possibility of sampling intrathoracic lymph nodes under ultrasound guidance (endosonography) through the airways (EBUS-TBNA) or the esophagus (EUS-FNA and EUS-B-FNA) [50].

### 3.1. Endobronchial Ultrasound-Guided Transbronchial Needle Aspiration (EBUS-TBNA)

EBUS-TBNA was introduced in 2002 to stage the mediastinum in lung cancer patients [51]. Over the last decade, it has also been useful for the diagnosis of lymphoma and benign conditions, such as tuberculosis and sarcoidosis [52].

EBUS-TBNA has revolutionized the sampling method because it is a minimally invasive procedure, performed with a flexible ultrasound bronchoscope with an integrated convex transducer (7.5 MHz) and Doppler mode that offers the operator a direct (real-time) visualization of lymph nodes and its characteristics, as well as surrounding structures, allowing the needle tip to be directed to the target area [11,23]. Once the target lesion is identified, transbronchial punctures are done, frequently, with a 22-gauge needle. A minimum of three to four needle passes are done for each lymph node. Then, aspirated specimens are expelled into a container with preservative liquid, and a cytoblocis prepared and stained for subsequent cytological examination. Sedation is performed according to institutional practice [53].

Convex-probe–EBUS enables access to the upper paratracheal (stations 2R and 2L), lower paratracheal (stations 4R and 4L), subcarinal (station 7), hilar (stations 10R and 10L) and interlobar (stations 11R and 11L) lymph nodes [50]. Its diagnostic yield for sarcoidosis ranges from 80 to 94% [16,22,23,24,25,26,27,28,29,54] and the pooled sensitivity for the diagnosis of hilar/mediastinal lesions was greater than 90% in patients with suspected/known lung cancer, and around 80% in those with sarcoidosis [23,38,39,55,56] (Figure 1). Ultrasound frequently reveals a typical pattern of isoechoic or hypoechoic lymph nodes, sometimes with prominent vessels and sometimes with a central hyperechoic strand within these nodes. However, these EUS characteristics are not distinctive of sarcoidosis, and cannot be used to distinguish these lesions from tuberculosis or malignancy [57].

Trisolini et al. [58] performed a systematic review and meta-analysis to determine the external validity of EBUS-TBNA for the diagnosis of sarcoidosis in non-selected populations. Fourteen studies were included and the results established an excellent diagnostic performance for EBUS-TBNA (pooled diagnostic yield 79%; sensitivity 84%), comparable to the results from reference centers that included populations with a very high prevalence of sarcoidosis [58].

According to Neves et al. [56], an adequate selection of patients is essential for the use of EBUS-TBNA in sarcoidosis, and the evaluation for further investigation should take into account the pre-test probability of sarcoidosis or alternative diagnosis, especially in stage I sarcoidosis. Thus, EBUS-TBNA, if available, should be routinely used in patients undergoing bronchoscopy for the diagnosis of sarcoidosis, especially in those with Scadding stage I or II [16].

### 3.2. Endoscopic Ultrasound-Guided Fine Needle Aspiration (EUS-FNA)

EUS-FNA is complementary to EBUS-TBNA. In brief, it is a real-time procedure, through the esophagus, providing access to stations 2L, 4L, 7, paraesophageal (station 8) and pulmonary ligament (station 9) lymph nodes [50]. It is not the first choice for sampling of paratracheal, hilar and interlobar nodes, especially on the right side, usually enlarged in patients with sarcoidosis [30]. Its use was first described in 2010, following a negative conventional bronchoscopy [33,59]. It is a highly sensitive, accurate, fast, safe and minimally-invasive method and its diagnostic yield for sarcoidosis varies between 77 and 94% [16,30,31,32,33].

Unlike EBUS-TBNA or cTBNA, additional procedures, such as TBLB or EBB, cannot be performed at the same time as EUS-FNA [60]. EUS-FNA is better tolerated than EBUS-TBNA, and is usually associated with fewer doses of anesthetics and sedatives, shorter procedure times and fewer oxygen desaturations [57]. It is also a good choice in cases of poor respiratory reserve and/or intractable cough [16].

In 2015, Gnass et al. [61] performed a randomized study comparing the diagnostic yield of cTBNA with EBUS-TBNA and EUS-FNA in patients clinically and radiologically suspected of sarcoidosis with mediastinal and hilar lymphadenopathies (stages I and II). The sensitivity and accuracy of cTBNA compared to EBUS-TBNA and EUS-FNA were, respectively, 62.5% and 64.7%, 79.3% and 80.0%, and 88.6% and 88.9%. The sensitivity and accuracy of EBUS-TBNA were higher (*p* = 0.139), as was that of EUS-FNA when compared with cTBNA (*p* = 0.012). In this study, no significant differences were observed between the sensitivity and accuracy of EUS-FNA and EBUS-TBNA.

A combined approach of EUS-FNA and EBUS-TBNA, already employed in some centers, could be a potential substitute for other invasive methods to evaluate uncertain mediastinal/hilar masses, as shown by some authors [22,29].

In 2017, Kocon et al. [62] published a prospective randomized study to assess the diagnostic yield of EBUS-TBNA and EUS-FNA, and to compare them with standard diagnostic techniques (such as EBB, TBLB and cTBNA). The authors stated that EUS-FNA had the highest diagnostic yield (75%) as a standalone test when compared with EBUS-TBNA (62%), but the combination of the two endosonographic techniques showed a diagnostic yield of 80%, whereas the combination of standard bronchoscopic procedures led to a diagnostic yield of 64%. However, EBUS-TBNA is generally performed by pulmonary physicians or thoracic surgeons, while EUS-FNA is performed primarily by gastroenterologists. This fact increases costs, as well as waiting times, for patients demanding both techniques [63].

### 3.3. Transesophageal Ultrasound-Guided Needle Aspiration with the Use of an Echo Bronchoscope (EUS-B-FNA)

In 2009, Hwangbo et al. [64] described the use of the echobronchoscope for transesophageal needle aspiration, called transesophageal ultrasound-guided needle aspiration with the use of an echo bronchoscope (EUS-B-FNA) [63,64,65].

Dhooria et al. [63] performed a systematic review and meta-analysis on the utility and safety of EUS-B-FNA in the diagnosis of mediastinal lymph node enlargement. In lung cancer staging, the sensitivity of the combined procedure was significantly higher than that of EBUS-TBNA alone (91% vs. 80%, *p* = 0.004), and no serious complications were reported. The authors concluded that EBUS-TBNA combined with EUS-B-FNA is an effective and safe approach, superior to EBUS-TBNA alone, in the diagnosis of mediastinal lymphadenopathies [63].

There are few studies on the application of this technique in the diagnosis of sarcoidosis. Oki et al. [32] used EUS-B-FNA in 33 patients for the diagnosis of stage I/II sarcoidosis, and achieved a diagnostic yield of 86% without complications. Nonetheless, there are multiple barriers to get started with EUS-B-FNA, but probably the most important ones are related to knowledge and training; thus, new operators should follow a structured training curriculum [65].

### 3.4. Transbronchial Lung Cryobiopsy (TBLC)

Transbronchial lung cryobiopsy (TBLC) is a recent and useful technique for the diagnosis of DPLD [66,67,68]. The procedure can be performed using a combination of rigid and flexible bronchoscopy under general anesthesia with manual jet ventilation (with a working pressure of approximately 2 bar). A flexible cryoprobe is introduced through the working channel and biopsies are taken under fluoroscopic guidance. Once brought into position, the probe is cooled to −85 °C with nitrogen oxide for approximately 5–6 s (3–4 s when sarcoidosis is the main suspicion), thus freezing the lung tissue in contact with the probe. A Fogarty balloon is prophylactically placed in the selected segmental bronchus and inflated immediately after the biopsy. The usual contraindications for TBLC are platelet count <70,000/mm^3^, international normalized ratio >1.5 or activated partial thromboplastin time >50 s, partial oxygen pressure in arterial blood (PaO2) <55 mmHg, forced expiratory capacity (FVC) <50%, forced expiratory volume in the first second (FEV1) <0.8 L, diffusing capacity for carbon monoxide (DLCO) <35%, diffuse bullous disease, heart disease or a transthoracic echocardiogram estimated pulmonary artery systolic pressure >40 mmHg [69].

TBLC diagnostic yield for undifferentiated interstitial lung disease varies from 74 to 98% when interpreted alone and with a pooled estimate of around 80% [70,71,72,73,74]. In relation to sarcoidosis, in 2017, Aragaki-Nakahodo et al. [34] performed a retrospective analysis in a tertiary-care of 36 patients with a suspected diagnosis of sarcoidosis, describing an overall diagnosis of 66.7% for TBLC.

TBLC overcomes the TBLB limitations with regards to quality and sample size, and it avoids more invasive modalities, such as surgical lung biopsy in atypical cases (e.g., those without marked adenopathies) and can be complementary to endosonography procedures, such as EBUS-TBNA [34].

The advantage of TBLC over TBLB is related to a larger biopsy size with preserved histopathology (avoiding the crush effect of TBLB) [16], which may facilitate, for example, the performance of the genetic profile for sarcoidosis as an upcoming diagnostic tool [75]. However, some authors argue against the use of TBLC as an alternative to TBLB to obtain lung tissue in cases of suspected sarcoidosis. Among others, they defend that in these cases, the combined diagnostic yield of EBB, TBLB and EBUS-TBNA is almost 100%, and there is no evidence to suggest that TBLC can improve this result. In addition, TBLB can be done with minimal sedation and with a flexible bronchoscope as an outpatient procedure, with fewer complications than TBLC, such as bleeding (1.3% vs. 15.6%), pneumothorax (5.5% vs. 10.9%) and mortality (0.2% vs. 3.0%), aside from being a less expensive procedure [34]. Moreover, in 2017, Almeida et al. [69] suggested that TBLC proficiency was achieved around the 70th procedure, underscoring the importance of experience by TBLC technicians.

In short, the data suggest that TBLC should be restricted to patients with suspected sarcoidosis and inconclusive TBLB, in whom an alternative diagnosis is likely. However, data on TBLC are still limited, as there are significant variations in technique [70,71,72] and complications rates [67]. Thus, further studies should be performed to clarify its role in the diagnosis of sarcoidosis. In addition, in cases of endoscopic’ inconclusive results, other techniques, such as CT-guided biopsy, should be considered, especially in cases of subpleural/peripheral abnormalities or even nodules/consolidations.

## 4. A Key Emphasis to EBUS-TBNA and Standard Bronchoscopic Modalities

On the whole, although the diagnostic superiority of EBUS-TBNA, as an autonomous modality over conventional bronchoscopic modalities, can be reasonably assumed by comparisons of pooled estimates in the literature, evidence-based data from adequately planned randomized clinical trials are still lacking [23]. Table 2 summarizes several studies published in the last decade, comparing EBUS-TBNA and EUS-FNA with conventional bronchoscopic techniques for the diagnosis of sarcoidosis.

In 2013, Bartheld et al. [37] showed a higher diagnostic yield for endosonography (EBUS-TBNA or EUS-FNA) (80%, 95% confidence interval (CI) = 73–86%) when compared with conventional bronchoscopy (TBLB and EBB) (53%, 95% CI = 45–61%) in patients with suspected pulmonary sarcoidosis (stages I and II). One year later, Gupta et al. [76] performed a randomized trial comparing the diagnostic yield of cTBNA versus EBUS-TBNA, with TBLB and EBB being executed in all cases. Alone, EBUS-TBNA had the highest yield (74.5%), which was significantly better than cTBNA (48.4%, *p* = 0.004) and EBB (36.3%, *p* < 0.0001), but not better than TBLB (69.6%, *p* = 0.54). The addition of EBB and TBLB to cTBNA led to improved diagnosis, whereas TBLB (but not EBB) significantly improved EBUS-TBNA yield. Thus, it was stated that EBUS-TBNA alone has the highest diagnostic yield in sarcoidosis, but it should be combined with TBLB for optimal results. With regards to cTBNA (plus EBB and TBLB), the diagnostic yield was similar to that of EBUS-TBNA + TBLB.

In 2016, Hu et al. [60] published a meta-analysis with 16 studies (only five RCTs), with a total of 1823 participants, who compared the efficacy of EBUS-TBNA with that of standard modalities in the diagnosis of sarcoidosis (the corresponding articles are marked with an asterisk in Table 2). The authors demonstrated that the absolute diagnostic yield of EBUS-TBNA + TBLB + EBB ranged from 86.4% to 100%, whereas that of cTBNA + TBLB + EBB ranged from 85.5% to 92.9%. Moreover, EBUS-TBNA was found to be more effective than TBLB for the diagnosis of stage I and II sarcoidosis (especially stage I), and more effective than TBLB + EBB, only for the diagnosis of stage I sarcoidosis [60].

In 2017, Bonifazi et al. [23] performed a randomized trial in a direct comparison between cTBNA and EBUS-TBNA for the diagnosis of hilar/mediastinal lesions. The authors randomized a total of 253 patients to either EBUS-TBNA (*n* = 127) or cTBNA (*n* = 126), and 31 patients from the cTBNA group subsequently performed EBUS-TBNA. The EBUS-TBNA sensitivity was higher, but not significantly higher than that of cTBNA (respectively, 92% (95% CI = 87–97) and 82% (95% CI = 75–90), *p* > 0.05). The sensitivity of the staged strategy was 94% (95% CI = 89–98). There were no major complications reported. Thus, the authors concluded that EBUS-TBNA was the best diagnostic tool, although not significantly superior to cTBNA. Additionally, the authors also aimed to assess the cost-effectiveness profile of a staged strategy, including cTBNA as the initial test followed by EBUS-TBNA, in the case of inconclusive rapid on-site evaluation (ROSE) results. These findings suggest that, in selected scenarios with high probability of success of the standard procedure, they can be included in a staged strategy with cTBNA as the initial test, followed by EBUS-TBNA, in cases of inconclusive results, without increasing the complications rate and reducing costs [23]. Thus, as previously mentioned by Mondoni et al. [11], these data suggest that, in selected cases and if EBUS-TBNA is not available, particularly in developing countries, clinicians should perform standard bronchoscopic techniques [60].

### 4.1. Complications of Endoscopic Ultrasound-Guided Techniques

A systematic review and meta-analysis of 15 studies by Agarwal et al. [52] investigated the diagnostic yield and safety of EBUS-TBNA in sarcoidosis. In short, they concluded that the procedure is safe—only five minor complications (pneumothorax, bleeding, airway edema/hypoxemia (*n* = 2), prolonged cough) were reported in the total of patients. Bartheld et al. [82] published a systematic review, including 16,181 patients (EBUS: 9119; EUS: 6042 and EBUS + EUS: 1020 patients) to determine the incidence and to identify the population at risk of serious adverse events (AE) attributable to these endosonographic diagnostic tools. Mortality was not reported and severe AE were very rare (0.14%), suggesting that endosonography is a safe technique for intrathoracic evaluation of the mediastinum. However, the true incidence of severe AE may be underestimated, possibly due to the low reporting of complications [82]. In addition, Hu et al. [60] in their meta-analysis (total of 1823 participants), found higher rates of complications for TBLB than for EBUS-TBNA, with these being specifically pneumothorax (*n* = 15) and bleeding (*n* = 8) after TBLB, and severe cough (*n* = 1) and minor bleeding (*n* = 4) after EBUS-TBNA. Also noteworthy are the infectious serious adverse events (mediastinitis and mediastinal abscess formation) of this technique in the subset of sarcoidosis patients [82,83]. Therefore, EBUS-FNA should be preferred and EUS-FNA, probably alongside antibiotic prophylaxis, should only be done after a non-diagnostic bronchoscopic work-up.

### 4.2. Factors Affecting Diagnostic Accuracy of Endoscopic Ultrasound-Guided Techniques

In the available literature, several factors have been stated as affecting the diagnostic accuracy of these techniques: type of lesion, size, location, experience of the endoscopist and cytopathologist, sampling pattern [84,85], needle sizes, number of passes, availability of ROSE [84] and economic status of the patient [19,84], among other factors. Data available concerning this issue are sometimes controversial.

#### 4.2.1. Size of Needles

Tremblay et al. [27] reported a higher diagnostic yield of EBUS-TBNA, in patients with mediastinal adenopathy and clinical suspicion of sarcoidosis, when performed with a 22-gauge needle versus cTBNA performed with a standard 19-gauge needle. The diagnostic yield was 53.8% for the cTBNA vs. 83.3% for EBUS-TBNA group. After a blinded research pathology review, the diagnostic yield was 73.1% vs. 95.8% in favor of the EBUS-TBNA group, an absolute increase of 22.7% (*p* = 0.05; 95% CI = 1.9–42.2) [27]. Therefore, proper needle guidance may be more relevant than a little enlargement of the sample size.

Muthu et al. [86] performed the largest prospective RCT, comparing the performance of EBUS-TBNA with a 21-gauge needle versus a 22-gauge needle, and concluded that the diagnostic yield in sarcoidosis was similar, regardless of the size of the aspiration needle. On the other hand, Tyan et al. [87] performed a first experiment with a flexible 19-gauge EBUS-TBNA needle and found a diagnostic yield of 93% for sarcoidosis, suggesting that a larger lumen could improve the likelihood of finding non-caseating granulomas. In addition, no complications occurred, except for a patient with moderate bleeding, which was subsequently successfully resolved [87].

#### 4.2.2. Rapid On-Site Evaluation (ROSE)

ROSE is a technique that ensures a proper samples’ handling and processing [60]. In 2007, Micames et al. [25], in a systematic review and meta-analysis, suggested that the accuracy of EUS-FNA is increased by ROSE. However, most of the literature related to EBUS-TBNA does not indicate a significant difference in its diagnostic yield according to ROSE availability [88,89,90,91,92].

In 2015, Trisolini et al. [58] stated that EBUS-TBNA with ROSE reduced the number of needle passes, the complication rate, procedural time, additional bronchoscopic procedures and even costs. In fact, Madan et al. [20], in a prospective randomized study, showed that when TBNA is performed in patients suspected of sarcoidosis, c-TBNA with ROSE and EBUS-TBNA (with or without ROSE) are superior to c-TBNA alone. However, they did not find significant differences in granuloma detection among the four groups. In addition, in 2017, Sehgal et al. [93] performed a systematic review and meta-analysis describing the usefulness of ROSE in individuals undergoing TBNA. Five studies (618 individuals; two EBUS-TBNA, two cTBNA and one using both techniques) were pointed out. There was no significant improvement in the diagnostic yield with ROSE. Thus, further studies are needed to clarify the real value of ROSE in reducing the need for additional bronchoscopic procedures.

#### 4.2.3. Size and Topography of Lymph Nodes

Data concerning the impact of the size and topography of lymph nodes in diagnostic accuracy of this technique are controversial. Some authors have shown that increased lymph node size supports an EBUS-based diagnosis of malignancy. In contrast, others have suggested that lymph node size has little effect on diagnostic accuracy or sensitivity of EBUS-TBNA. Abu-Hijleh et al. [94] found that samples taken from lymph nodes ≥20 mm in size always contained sufficient material for analysis. The same authors reported that samplings of the 4R, 4L, 7, 10/11R and 10/11L lymph nodes did not differ in terms of diagnostic accuracy. Cetinkaya et al. [95] showed the sensitivity of EBUS-TBNA was lower in the region of lymph node 4L than in those of the other lymph nodes. The authors reported that the overall diagnostic accuracy of EBUS-TBNA was not affected by the number of lymph nodes sampled, the number of times a lymph node region was sampled or lymph node size.

#### 4.2.4. Endoscopist Experience

Guidelines on training requirements for EBUS-TBNA, based on expert opinion, recommend at least 40–50 procedures for initial proficiency acquisition [96,97]. However, emerging evidence suggests that simulation-based training may overcome the initial steep portion of the learning curve of EBUS [98,99]. Sehgal et al. [100] suggested that simulator-based training should be performed as well as the traditional training methods. As well as this, this systematic review found that the number needed to overcome the initial learning curve for EBUS is highly variable. However, it is likely that to achieve an accuracy of at least 80%, about 37–44 procedures are required. Finally, computer simulator and assessment tools, such as EBUS Skill and Task Assessment Tool, can objectively evaluate the outcome of EBUS training [100].

#### 4.2.5. Number of Passes

Usually, for EBUS-TBNA and EUS-FNA, three to four aspirations per lymph node station are recommended, as the diagnostic yield curve reaches its plateau after three samples [99,101,102]. With ROSE, a smaller number of samples may be possible, as the procedure is stopped once a diagnosis of malignancy, granulomatous or inflammatory condition is reached [43].

## 5. Conclusions

In overall, there are several bronchoscopic techniques available to obtain tissue samples for sarcoidosis diagnosis, although most of the evidence considers the diagnostic superiority of endosonographic techniques, such as EBUS-TBNA alone or in combination with EUS-FNA, over conventional bronchoscopic modalities, for Scadding stages I and II of the disease. The utility of endosonography procedures in the diagnosis of sarcoidosis cannot be neglected, but its usefulness as the only diagnostic procedure remains a challenge with concerns to the other stages of the disease. Though it is not always significantly superior to the standard bronchoscopic techniques, for optimal performance, the combined approach with TBLB and EBB is recommended. Moreover, TBLC may also have a role in sarcoidosis diagnosis, overcoming TBLB limitations, avoiding more invasive modalities and being complementary to endosonography procedures, such as EBUS-TBNA. In this context, the following diagnostic flowchart is proposed for bronchoscopic diagnosis of sarcoidosis (Figure 2).

## Figures and Tables

**Figure 1 jcm-08-01327-f001:**
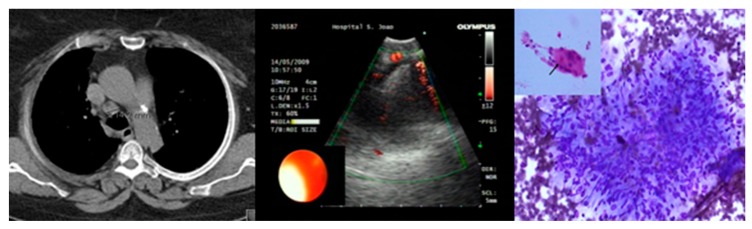
Endobronchial ultrasound in a patient with mediastinal lymph node abnormalities and suspicion of sarcoidosis diagnosis. EBUS-TBNA of an enlarged lymph node (station 4R) with homogeneous texture showed a non-necrotizing granuloma (hematoxylin-eosin and Giemsa).

**Figure 2 jcm-08-01327-f002:**
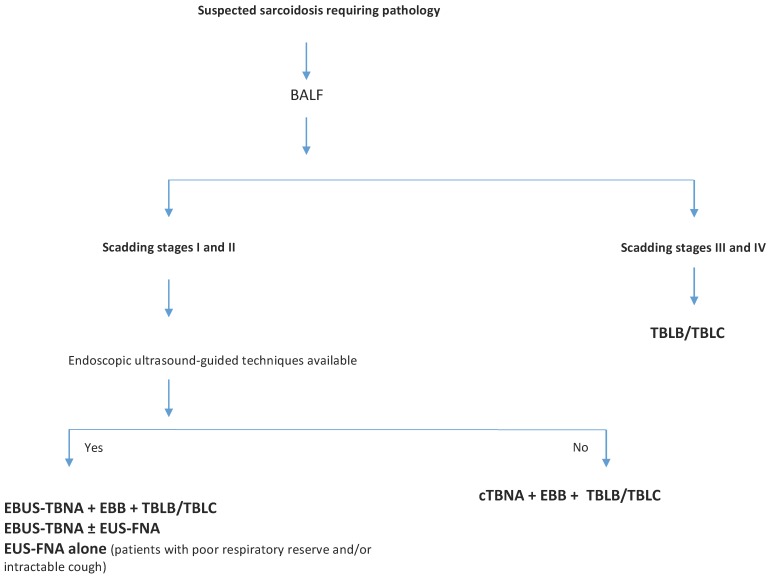
Proposed diagnostic flowchart for bronchoscopic diagnosis of sarcoidosis. BALF: broncolaveolar lavage fluid; EBUS-TBNA: endobronchial ultrasound-guided transbronchial needle aspiration; EUS-FNA: endoscopic ultrasound-guided fine needle aspiration; EBB: endobronchial biopsy; TBLB: transbronchial lung biopsy; cTBNA: conventional transbronchial needle aspiration; TBLC: transbronchial lung cryobiopsy.

**Table 1 jcm-08-01327-t001:** Bronchoscopic techniques to diagnose sarcoidosis and their overall diagnostic yield.

	Techniques	Diagnostic Yield	References
Standard	EBB	20–61%	[15,16]
	TBLB	37–90%	[16,17,18]
	cTBNA	6–90%	[16,19,20,21]
Recent	EBUS-TBNA	80–94%	[16,22,23,24,25,26,27,28,29]
	EUS-FNA	77–94%	[16,30,31,32,33]
	EUS-B-FNA	86% (*)	[32]
	TBLC	66.7% (*)	[34]

cTBNA, conventional transbronchial needle aspiration; EBB, endobronchial biopsy; EBUS-TBNA, endobronchial ultrasound-guided transbronchial needle aspiration; EUS-FNA, endoscopic ultrasound-guided fine needle aspiration; EUS-B-FNA, transesophageal ultrasound-guided needle aspiration with the use of an echo bronchoscope; TBLB, transbronchial lung biopsy; TBLC, transbronchial lung cryobiopsy. (*) the only available data.

**Table 2 jcm-08-01327-t002:** Diagnostic yield for sarcoidosis of the different bronchoscopic modalities among the distinct studies (adapted from Hu et al. [60]).

Study/Year	Scadding Stage	Number of Patients, *n*	EBUS-TBNA (%)	TBNA (%)	TBLB (%)	EBB (%)	TBLB + EBB (%)	EBUS-TBNA or EUS-FNA	EBUS-TBNA Group (%)	TBNA Group (%)	EUS-B-FNA	EBUS-TBNA + EUS-FNA
RCT												
Tremblay et al., 2009 * [27]	I or II	50	83.3	60.9	-	-	-	-	-	-	-	-
Bartheld et al., 2013 * [37]	I, II, III	304		-	-	-	53.0	80.0	-	-	-	-
Gupta et al., 2014 * [77]	I or II	117	74.5	48.4	69.6	36.3	-	-	92.7	85.5	-	-
Li and Jiang, 2014 * [26]	I or II	57	93.0	64.0	36.4	5.0	-	-	-	92.9	-	-
Gnass et al., 2015 * [61]	I or II	64	76.7	58.8	-	-	-	-	-	-	86.1	-
Prospective												
Navani et al., 2011 * [78]	I or II	27	85.0	-	29.6	11.1	33.3	-	92.6	-	-	-
Oki et al., 2012 * [17]	I or II	54	94.0	-	37.0	-	-	-	-	-	-	-
Plit et al., 2013 * [79]	I or II	49	91.8	-	67.3	28.6	-	-	-	-	-	-
Goyal et al., 2014 * [15]	I, II, III, IV	151	57.1	22.4	68.7	49.6	81.4	-	86.4	86.9	-	-
Kocón et al., 2017 [62]	I or II	100	61.9	44.0	42.0	12.0	-	-	-	-	74.55	80.0
Retrospective												
Nakajima et al., 2009 * [28]	I or II	35	91.4	-	40.0	-	-	-	-	-	-	-
Zhang et al., 2011 * [80]	I, II, III	50	86.7	82.5	-	-	66.7	-	-	-	-	-
Plit et al., 2012 * [18]	I or II	37	84.0	-	78.0	27.0	-	-	100.0	-	-	-
Hong et al., 2013 * [81]	I or II	31	90.0	-	35.0	6.0	39.0	-	94.0	-	-	-
Dziedzic et al., 2015 * [35]	I or II	653	84.0	-	43.9	29.7	54.0	-	89.0	-	-	-

EBUS-TBNA: endobronchial ultrasound-guided transbronchial needle aspiration; TBNA: transbronchial needle aspiration; TBLB: transbronchial lung biopsy; EBB: endobronchial biopsy; EBUS-TBNA group: EBUS TBNA + TBLB + EBB; TBNA-group: TBNA + TBLB + EBB; EUS-FNA: endoscopic ultrasound-guided fine needle aspiration. RCT: randomized controlled trial; -: no data; %: diagnostic yield of sarcoidosis; *, studies comparing the efficacy of EBUS-TBNA with that of standard modalities in the diagnosis of sarcoidosis.

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
