# Peer review of "Role of Bronchoscopic Techniques in the Diagnosis of Thoracic Sarcoidosis"

_jcm, 2019, doi:10.3390/jcm8091327_

Round 1

Reviewer 1 Report

Overall content is well done.

Recommend review and edits of grammar and writing style.

Sample images of TBLB, TBLC, EBUS TBNA, mucosal irregularities, etc may be of interest to readers.

Please find below a few suggestions.

17 inflammation. Currently, a proper diagnosis, with a high level of confidence, is conceived as the key --> 

17 inflammation. Currently, a proper diagnosis, with a high level of confidence, is considered as key

18 to an appropriate clinical and therapeutic management of the disease. In this sense, this review aims-->

18 to appropriate diagnosis and management of the disease. In this sense, this review aims

20 drifting from the main classical endoscopic techniques [endobronchial biopsy (EBB), transbronchial 

21 lung biopsy (TBLB), conventional transbronchial needle aspiration (cTBNA) and bronchoalveolar

22 lavage fluid (BALF)] to recent advances [endobronchial ultrasound-guided transbronchial needle

23 aspiration (EBUS-TBNA), endoscopic ultrasound-guided fine-needle aspiration (EUS-FNA),

24 transesophageal endobronchial ultrasound-guided needle aspiration (EUS-B-NA) and

25 transbronchial lung cryobiopsy (TBLC)] in this field. -->

20 incorporating newer techniques to established including endobronchial ultrasound-guided transbronchial needle aspiration (EBUS-TBNA), endoscopic ultrasound-guided fine-needle aspiration (EUS-FNA), transesophageal endobronchial ultrasound-guided needle aspiration (EUS-B-NA) and transbronchial lung cryobiopsy (TBLC). 

28 disease. Moreover, TBLC, overcoming TBLB limitations, avoiding more invasive modalities and

29 being complementary to endosonographic procedures, such as EBUS-TBNA, can be considered a

30 useful and safe diagnostic tool for thoracic sarcoidosis. -->

28 disease. Moreover, TBLC may be considered a useful and safe diagnostic tool for thoracic sarcoidosis, overcoming some limitations of TBLB, avoiding more invasive modalities and being complementary to endosonographic procedures, such as EBUS-TBNA.

50 Although it can affect any organ, the pulmonary and mediastinal lymph node involvement appear-->

50 Although it can affect any organ, pulmonary and mediastinal lymph node involvement appears

60 disease that courses with granulomas, such as tuberculosis, sarcoid-like reactions in cancers and -->

60 disease associated with granulomas, such as tuberculosis, sarcoid-like reactions in cancers and

64 namely pulmonary parenchyma and hilar lymph nodes abnormalities. The Scadding staging, which -->

64 namely pulmonary parenchymal and hilar lymph node abnormalities. The Scadding staging, which

70 caseating granulomas remains critical in diagnosing sarcoidosis. Moreover, the diagnosis of high -->

70 caseating granulomas remains critical in diagnosing sarcoidosis. Moreover, a diagnosis of high

74 selection depends on clinician’s pre-test confidence level for the diagnosis of sarcoidosis and the -->

74 of choice depends on clinician’s pre-test probability for the diagnosis of sarcoidosis and the

Author Response

Overall content is well done.

Answer: Thank you for the overall appreciation of our work

Recommend review and edits of grammar and writing style.

Answer: the whole manuscript was carefully revised

Sample images of TBLB, TBLC, EBUS TBNA, mucosal irregularities, etc may be of interest to readers.

Answer: According to the reviewer suggestion, a proper image was included.

Please find below a few suggestions.

17 inflammation. Currently, a proper diagnosis, with a high level of confidence, is conceived as the key -->

17 inflammation. Currently, a proper diagnosis, with a high level of confidence, is considered as key

Answer: Revised

18 to an appropriate clinical and therapeutic management of the disease. In this sense, this review aims-->

18 to appropriate diagnosis and management of the disease. In this sense, this review aims

Answer: Revised

20 drifting from the main classical endoscopic techniques [endobronchial biopsy (EBB), transbronchial

21 lung biopsy (TBLB), conventional transbronchial needle aspiration (cTBNA) and bronchoalveolar

22 lavage fluid (BALF)] to recent advances [endobronchial ultrasound-guided transbronchial needle

23 aspiration (EBUS-TBNA), endoscopic ultrasound-guided fine-needle aspiration (EUS-FNA),

24 transesophageal endobronchial ultrasound-guided needle aspiration (EUS-B-NA) and

25 transbronchial lung cryobiopsy (TBLC)] in this field. -->

20 incorporating newer techniques to established including endobronchial ultrasound-guided transbronchial needle aspiration (EBUS-TBNA), endoscopic ultrasound-guided fine-needle aspiration (EUS-FNA), transesophageal endobronchial ultrasound-guided needle aspiration (EUS-B-NA) and transbronchial lung cryobiopsy (TBLC).

Answer: Revised

28 disease. Moreover, TBLC, overcoming TBLB limitations, avoiding more invasive modalities and

29 being complementary to endosonographic procedures, such as EBUS-TBNA, can be considered a

30 useful and safe diagnostic tool for thoracic sarcoidosis. -->

28 disease. Moreover, TBLC may be considered a useful and safe diagnostic tool for thoracic sarcoidosis, overcoming some limitations of TBLB, avoiding more invasive modalities and being complementary to endosonographic procedures, such as EBUS-TBNA.

Answer: Revised

50 Although it can affect any organ, the pulmonary and mediastinal lymph node involvement appear-->

50 Although it can affect any organ, pulmonary and mediastinal lymph node involvement appears

Answer: Revised

60 disease that courses with granulomas, such as tuberculosis, sarcoid-like reactions in cancers and -->

60 disease associated with granulomas, such as tuberculosis, sarcoid-like reactions in cancers and

Answer: Revised

64 namely pulmonary parenchyma and hilar lymph nodes abnormalities. The Scadding staging, which -->

64 namely pulmonary parenchymal and hilar lymph node abnormalities. The Scadding staging, which

Answer: Revised

70 caseating granulomas remains critical in diagnosing sarcoidosis. Moreover, the diagnosis of high -->

70 caseating granulomas remains critical in diagnosing sarcoidosis. Moreover, a diagnosis of high

Answer: Revised

74 selection depends on clinician’s pre-test confidence level for the diagnosis of sarcoidosis and the -->

74 of choice depends on clinician’s pre-test probability for the diagnosis of sarcoidosis and the

Answer: Revised

Reviewer 2 Report

This is a systematic review on the different bronchoscopic techniques that can be employed for the diagnosis of sarcoidosis.

Even if this paper doesn't allow any new information, the review is complete and well written.

Just few minor comments:

- maybe the introduction is too long (some generic consideration about sarcoidosis, such as epidemiological data, are out of the scope of this review)

- the Authors describe the technique of transesophageal ultrasound guided needle aspiration using an echo bronchoscope as "transesophageal endobronchial ultrasound guided needle aspiration" and for this technique they use the acronym of "EUS-B-NA". The definition is not correct since the adjective "endobronchial" is not adequate. It should be used the definition "transesophageal ultrasound guided needle aspiration with the use of an echo bronchoscope" and the acronym generally used in the literature is "EUS-B-FNA". Please, correct this point in the paper and tables;

-lines 141-143: it is reported that the lower diagnostic yield of cTBNA "may be associated with the mobility of mediastinal lymph nodes and the variable distance between the carina and the lymph node" and the reference to support this statement is Cameron et al, Cytopathology 2010. However, the paper of Cameron reports this statement from the original paper of Piet et al (Chest 2007; 131: 1783) where just very small lymph nodes were evaluated (1.2 cm). This statement did not find any other confirmation on larger series and on larger lymph nodes and it is not widely accepted as possible cause of lower diagnostic yield of cTBNA. I would avoid this affirmation.

- line 147: the Authors write that cTBNA is underused due to "undefined safety concerns". There are a lot of papers in the literature that established the safety of cTBNA. 

- line 203: the subcarinal lymph node station is 7, and not 7L

- line 265: concerning cryobiopsy, is reported a mortality of 3% in comparison to 0.2% of conventional transbronchial lung biopsy. 3% of mortality for cryobiopsy is not reported in the literature.

-Complications of endoscopic ultrasound-guided techniques. There are several reports of increased risk of mediastinitis using transesophageal approach in sarcoidosis. This complication is missed and should be reported and discussed, because it could be a reason to use as a first step transbronchial approach as suggested by some Authors.

Author Response

This is a systematic review on the different bronchoscopic techniques that can be employed for the diagnosis of sarcoidosis.

Even if this paper doesn't allow any new information, the review is complete and well written.

Answer: Thank you for the overall appreciation of our work

Just few minor comments:

- maybe the introduction is too long (some generic consideration about sarcoidosis, such as epidemiological data, are out of the scope of this review)

Answer: Introduction was restructured and shortened.

- the Authors describe the technique of transesophageal ultrasound guided needle aspiration using an echo bronchoscope as "transesophageal endobronchial ultrasound guided needle aspiration" and for this technique they use the acronym of "EUS-B-NA". The definition is not correct since the adjective "endobronchial" is not adequate. It should be used the definition "transesophageal ultrasound guided needle aspiration with the use of an echo bronchoscope" and the acronym generally used in the literature is "EUS-B-FNA". Please, correct this point in the paper and tables;

Answer: Revised

-lines 141-143: it is reported that the lower diagnostic yield of cTBNA "may be associated with the mobility of mediastinal lymph nodes and the variable distance between the carina and the lymph node" and the reference to support this statement is Cameron et al, Cytopathology 2010. However, the paper of Cameron reports this statement from the original paper of Piet et al (Chest 2007; 131: 1783) where just very small lymph nodes were evaluated (1.2 cm). This statement did not find any other confirmation on larger series and on larger lymph nodes and it is not widely accepted as possible cause of lower diagnostic yield of cTBNA. I would avoid this affirmation.

            Answer: This sentence was removed.

- line 147: the Authors write that cTBNA is underused due to "undefined safety concerns". There are a lot of papers in the literature that established the safety of cTBNA.

Answer: The correct word would be unconfirmed. This aspect was revised in the current version of the document.

- line 203: the subcarinal lymph node station is 7, and not 7L

Answer: Revised

- line 265: concerning cryobiopsy, is reported a mortality of 3% in comparison to 0.2% of conventional transbronchial lung biopsy. 3% of mortality for cryobiopsy is not reported in the literature.

Answer: Thank you for the reviewer advice. However, this information was based into the work published by Rajagopala S.Sarcoidosis: is cryobiopsy not cool enough?Lancet Respir Med. 2018 Sep;6(9):e44. doi: 10.1016/S2213-2600(18)30215-7. Epub 2018 Jul 11., which was based also in the previous work of Dhooria S, Mehta RM, Srinivasan A, et al. The safety and efficacy of different methods for obtaining transbronchial lung cryobiopsy in diffuse lung diseases. Clin Respir J 2018; 12: 1711–20.In the former document, the authors stated that:

Retrospective studies suggest that the diagnostic yield of TBLC in suspected sarcoidosis is similar to that of TBLB, at around 66·7%.However, TBLB can be done with minimal sedation and with a flexible bronchoscope alone as an outpatient procedure, with bleeding occurring in 1·3% of cases, pneumothorax in 5·5%, and mortality in 0·2%.By contrast, TBLC is typically done as an inpatient procedure under general anaesthesia, using a rigid bronchoscope or conduit and a cryoprobe with a blocker to tamponade severe bleeding. Bleeding occurs in 15·6%, pneumothorax in 10·9%, and mortality in 3% of patients who undergo TBLC”

 -Complications of endoscopic ultrasound-guided techniques. There are several reports of increased risk of mediastinitis using transesophageal approach in sarcoidosis. This complication is missed and should be reported and discussed, because it could be a reason to use as a first step transbronchial approach as suggested by some Authors.

Answer: The requested information by the reviewer was carefully included in the revised version of the manuscript.

Reviewer 3 Report

The authors perform a thorough and concise review of the pertinent literature regarding the diagnostic procedures available for thoracic sarcoidosis as well as the yield of each of these modalities. The authors trace the evolution from the conventional procedures used for diagnostic purposes to the more advanced bronchoscopic and endoscopic procedures available as well as more recently to the use of transbronchial cryobiopsy. The authors include a review of the pertinent pivotal clinical trials and other case series/studies supporting the diagnostic evolution from conventional bronchoscopic procedures to more advanced. This review will be helpful in informing general physicians, pulmonologists, and proceduralists who encounter patients with suspected pulmonary and/or thoracic sarcoidosis in determining which procedures are most beneficial for use in diagnosis and what modalities should be used in the sequential evaluation of these patients. 

The paper would benefit from a further expansion of section 4.2 and only needle size and ROSE are further elaborated upon. Additional information commenting on the effect of type of lesion, size, location, and experience of the endoscopist and cytopathologist, sampling pattern, and number of passes would be informative for the readers. 

Similarly, I would recommend clarifying the conclusion to suggest a specific diagnostic approach/algorithm that the authors would endorse when encountering patients with suspected thoracic sarcoidosis. This would be helpful for providers encountering these patients in the future and will inform their decision making. A figure representing the suggested algorithm would also be impactful.  

Other comments would be that the references seems to be out of order. I would check with the journal format to ensure that references are in sequential order. 

Overall, the manuscript would benefit from editing/review for grammar and syntax by an English editor/reviewer. There are also times where the diction changes from formal to very informal or colloquial. While these are not errors of content, they are numerous and throughout the paper to the point where correction would make the paper read and flow better. 

Author Response

The authors perform a thorough and concise review of the pertinent literature regarding the diagnostic procedures available for thoracic sarcoidosis as well as the yield of each of these modalities. The authors trace the evolution from the conventional procedures used for diagnostic purposes to the more advanced bronchoscopic and endoscopic procedures available as well as more recently to the use of transbronchial cryobiopsy. The authors include a review of the pertinent pivotal clinical trials and other case series/studies supporting the diagnostic evolution from conventional bronchoscopic procedures to more advanced. This review will be helpful in informing general physicians, pulmonologists, and proceduralists who encounter patients with suspected pulmonary and/or thoracic sarcoidosis in determining which procedures are most beneficial for use in diagnosis and what modalities should be used in the sequential evaluation of these patients.

Answer: Thank you for the overall appreciation of our work.

The paper would benefit from a further expansion of section 4.2 and only needle size and ROSE are further elaborated upon. Additional information commenting on the effect of type of lesion, size, location, and experience of the endoscopist and cytopathologist, sampling pattern, and number of passes would be informative for the readers.

Answer: The requested aspects by the reviewer were carefully included in the revised version of the manuscript.

Similarly, I would recommend clarifying the conclusion to suggest a specific diagnostic approach/algorithm that the authors would endorse when encountering patients with suspected thoracic sarcoidosis. This would be helpful for providers encountering these patients in the future and will inform their decision making. A figure representing the suggested algorithm would also be impactful. 

Answer: Conclusion section was revised accordingly. According to the reviewer suggestion a flowchart was also included.

Other comments would be that the references seems to be out of order. I would check with the journal format to ensure that references are in sequential order.

Answer: All references were updated.

Overall, the manuscript would benefit from editing/review for grammar and syntax by an English editor/reviewer. There are also times where the diction changes from formal to very informal or colloquial. While these are not errors of content, they are numerous and throughout the paper to the point where correction would make the paper read and flow better.

Answer: The whole manuscript was carefully revised by a native speaker.